# A reinforcement learning and sequential sampling model constrained by gaze data

William M. Hayes ⓘ *, Melanie J. Touchard

Psychology Department, Binghamton University State University of New York, Binghamton, New York, United States of America

* whayes2@binghamton.edu

## Abstract

Reinforcement learning models can be combined with sequential sampling models to fit choice-RT data. The combined models, known as RL-SSMs, explain a wide range of choice-RT patterns in repeated decision tasks. The present study shows how constraining an RL-SSM with eye gaze data can further enhance its predictive ability. Our model allows learned option values and relative gaze to jointly influence the accumulation of evidence prior to choice. We evaluate the model on data from two eye-tracking experiments (total N = 133) and test several variants of the model that assume different mechanisms for integrating values and gaze at the decision stage. Further, we show that it captures a variety of empirical effects, including gaze biases on choice and response time, as well as individual differences in absolute versus relative valuation. The model can be used to understand how learned option values interact with visual attention to influence choice, joining together two major (but mostly separate) modeling traditions.

### Author summary

When people are deciding between options they have encountered in the past, their preferences are largely based on their memory of past experiences with those options. However, simply looking at an option longer can increase the likelihood that it will be chosen, regardless of how rewarding the option is to the decision maker. Most theories of experience-based decision making do not account for this gaze effect. We introduce a tractable and scalable computational model that can predict choices and response times in repeated decision tasks while simultaneously accounting for gaze effects. The model can be used to understand the interplay of learning and visual attention in experience-based decision making.

**Data availability statement:** Data and code are available at https://github.com/william-hayes/RL-LBA-gaze-model.

**Funding:** The author(s) received no specific funding for this work.

**Competing interests:** The authors have declared that no competing interests exist.

## Introduction

Whether deciding what to eat for lunch, which route to take to avoid traffic, or who to trust for financial advice, many of our decisions require us to rely on information learned through experience. Our preferences in these situations are often influenced by the outcomes of previous choices. However, several other factors can influence these decisions, including the time spent deliberating and the amount of attention allocated to the available alternatives at the time of choice [1,2].

Traditionally, decisions from experience have been studied using reinforcement learning models (RL), which describe how decision makers adjust their choice behavior in response to feedback from the environment to maximize reward [3]. Most RL models use a static choice rule (softmax) to select actions based on expected values [4]. This choice rule has key limitations. First, it does not account for choice response times (RTs), which are known to reflect decision difficulty and speed-accuracy tradeoffs [5,6]. For example, it may take longer to make a decision if the available options are subjectively similar, or if the decision maker wants to find the best possible option as opposed to one that is just good enough. Second, in its usual form, the static choice rule in RL models does not account for attentional biases in action selection [7] (but see [8,9]). A decision maker may choose a less desirable option—based on their previous experience—simply because they spend more time fixating on it. Such effects would be missed by standard RL models.

Meanwhile, separate lines of research on perceptual and value-based decision making have led to the development of sequential sampling models (SSMs) that can simultaneously account for choices and RTs [10–13], as well as attentional biases [14,15]. Though exact implementations vary, SSMs generally assume that choices occur through a dynamic process in which evidence (or preference) for choosing a particular option accumulates over time until a decision threshold is reached. The time taken to reach the threshold, plus a small amount of time for stimulus encoding and response execution, determines the RT. In value-based decisions, the speed of evidence accumulation, or drift rate, for a particular option reflects its subjective value to the decision maker (or its relative subjective value). The height of the decision threshold reflects the decision maker's response caution, or the amount of evidence they require to make a choice.

SSMs have also been used to model the role of overt visual attention, or gaze, in value-based decisions [16,17]. This line of work has demonstrated robust gaze biases. In particular, the longer an option is fixated, the more likely it is to be chosen [14,15,18–21]. This effect persists after controlling for the subjective values of the choice options, and therefore cannot be explained away as merely a tendency to look longer at high-value options. Rather, gaze appears to either amplify [22] or add to [19,23] the value of an option during evidence accumulation. For example, in the attentional drift diffusion model (aDDM; [14]), looking at an option temporarily biases the drift rate toward the fixated option by discounting the value of the unfixated option: a value-amplifying effect. Other models assume that gaze adds a fixed bonus to the drift rate that is independent of the option value [19]. Which mechanism is best—additive or multiplicative—may depend on the choice task [22].

In summary, while RL models can explain how preferences are learned through trial-and-error, SSMs give a more complete account of the decision stage, incorporating choice RTs and (optionally) gaze biases. Fortunately, the two approaches can be combined by substituting an SSM for the static choice rule in an RL model [24–29]; for a review, see [1]. In this approach, learning occurs over trials according to an incremental RL mechanism, and choices are made through a sequential sampling process, with the drift rate(s) depending on the current learned values for each option. These combined RL-SSMs explain a variety of behavioral patterns in decisions from experience, including the effects of learning and choice difficulty on accuracy and response time [26]. However, to our knowledge, there is currently no RL-SSM that accounts for gaze biases and how they interact with value learning to influence behavior.

In the present study, we introduce an RL-SSM that can account for trial-and-error learning, choice RTs, and attentional biases in decisions from experience. The model explains a variety of empirical effects: Increasing accuracy and faster RTs over the course of learning, individual differences in overall accuracy and decision speed, and (in Experiment 2) context-dependent valuation [30–33]. When constrained with eye-tracking data, it also captures gaze effects that would be missed by existing RL-SSMs: For example, the longer an option is fixated relative to other options, the more likely it is to be chosen. The model has several practical advantages: it is lightweight, analytically tractable, and readily scalable to more complex tasks. In the following, we briefly describe the experiments that were used to evaluate the model before turning to a more detailed description of the model itself.

## Results

### Overview of experiments

We evaluated the model on data from two eye-tracking experiments (Fig 1A). Although both experiments were originally designed to examine context-dependent valuation in RL, the resulting data sets provide a rich testbed for evaluating RL-SSMs.

The task in the first experiment (N = 83) was adapted from a prior study [31]. Participants encountered several choices between eight symbols grouped in two sets of four. The symbol groupings, or contexts, remained fixed across the 60 learning phase trials. Every trial began with the presentation of the four symbols from a randomly selected context, but only two of the four were available to choose (indicated by asterisks). After making a choice, participants received probabilistic reward feedback from all four symbols, including the ones that they could not select. The rewards were point values drawn from Gaussian distributions with different expected values (EVs) and rounded to whole numbers. The instructed goal was to learn which symbols had the highest value and to maximize the total number of points earned.

The task in the second experiment (N = 50) was adapted from another prior study [34], and parts of the data have been reported elsewhere [35]. In this experiment, the symbols were grouped in four 2-option contexts instead of two 4-option contexts, the learning phase was longer, and the task ended with a transfer test in which participants encountered choices between all possible pairs of symbols without receiving feedback. The transfer test was designed to reveal whether participants had learned the absolute EVs of the symbols (context-independent) or the relative values (context-dependent). In the latter case, we should observe a predictable pattern of suboptimal preferences in the transfer test. For example, given a choice between options B and C, absolute values would dictate choosing C because it has a higher absolute EV (21 compared to 18). Relative values, in contrast, would dictate choosing B because it was better locally in its original context.

The trials were self-paced in both experiments, which allowed us to measure choice RTs. Importantly, because the choice contexts were presented in a random order, participants could not anticipate which set of symbols they would see on the next trial and decide beforehand which symbol to choose [9,36]. This is crucial for our modeling assumption that gaze directly biases the evidence accumulation process leading up to choice. Next, we introduce the gaze-constrained RL-SSM that we used to fit choice-RT data in both experiments.

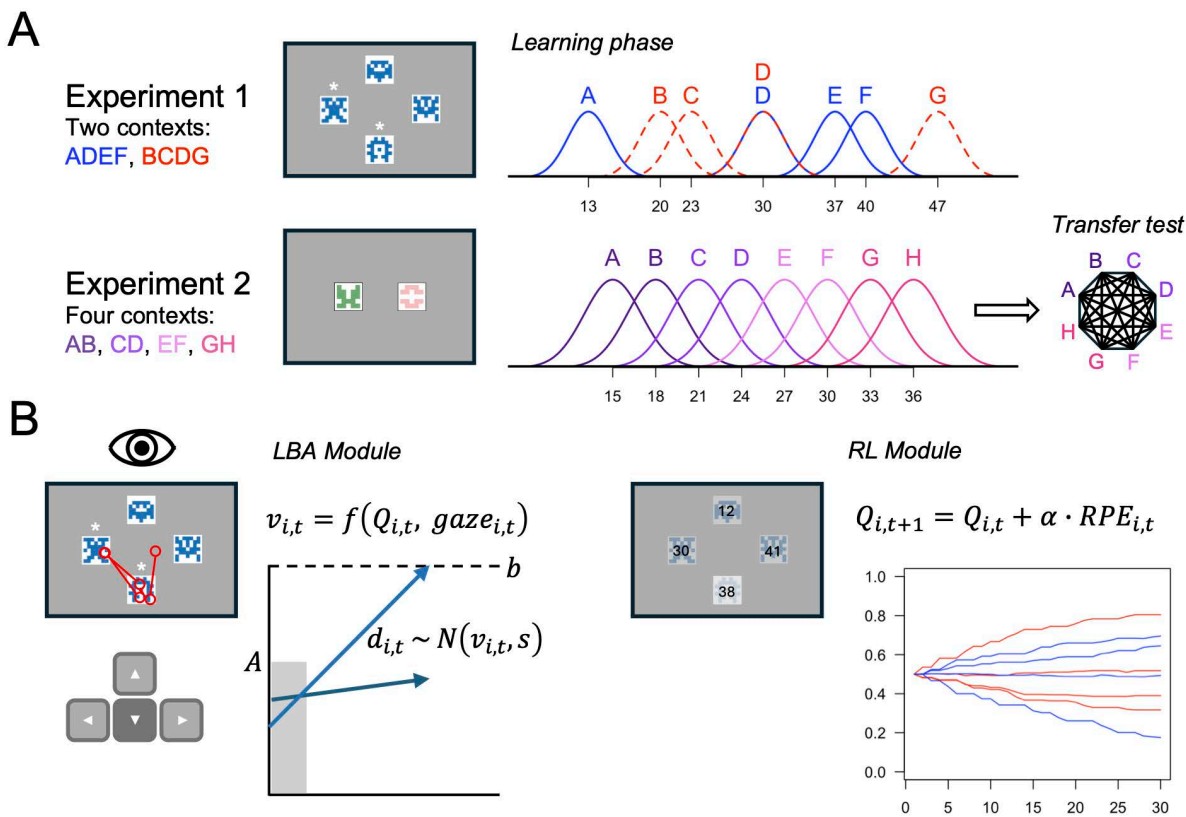

**Fig 1. Overview of experiments and model. (A)** In the learning phase, symbols were encountered in fixed groupings (contexts). On every trial, the symbols from a randomly selected context were presented on screen. In Experiment 1, only two of the four symbols were available to choose (indicated with asterisks). The available symbols varied from trial to trial. Choices were followed by probabilistic reward feedback from all options seen on that trial. Experiment 2 included a transfer test after the learning phase that involved choices between all possible pairs of symbols without feedback. The outcome distributions in both experiments were Gaussian with option-specific means (SD = 2). **(B)** Our model combines separate modules for decision making and learning from choice feedback. The LBA module uses linear ballistic accumulation to make decisions. The mean drift rates ($v_{i,t}$) are assumed to be a function of the current estimated values of the available symbols ($Q_{i,t}$) and the proportional gaze allocated to each symbol during the time interval between trial onset and choice ($gaze_{i,t}$). The RL module updates the estimated value of each symbol in response to feedback using a prediction error-driven learning rule (RPE = reward prediction error). Symbols adapted from https://github.com/sophiebavard/online_task.

## Computational model

Our computational model combines a linear ballistic accumulation (LBA) module for decision making [13] and an RL module for learning the values of options through experience (Fig 1B). The RL module maintains estimated values for each option that it updates in response to feedback from the environment. At choice time, the current value estimates for the available options are mapped to the drift rates in the LBA module, which determine how the evidence accumulation process unfolds. The model assumes that drift rates are also influenced by trial-to-trial fluctuations in visual attention to the available options, which allows it to explain a wider range of behavioral effects compared to previous RL-SSMs [25–28].

The LBA module assumes that choices result from a dynamic sequential sampling process. Evidence (or preference) accumulates in a linear trajectory toward the decision threshold ($b$), separately for each of the available options. The chosen option is the one whose accumulator reaches the threshold first. The time taken for the winning accumulator to reach the threshold, plus a small amount of nondecision time ($t_0$), determines the choice RT. There are two sources of between-trial variability in the LBA module. First, the starting point for each option's accumulator is assumed to be drawn

independently on each trial from a $\mathcal{U}(0, A)$ distribution with lower bound 0 and upper bound $A$. Second, the drift rate, or slope, of the $i$th option's accumulator on trial $t$, $d_{i,t}$, is assumed to be drawn independently from a $\mathcal{N}(v_{i,t}, s)$ distribution with mean $v_{i,t}$ and standard deviation $s$.

In our model, the mean of the drift rate distribution for the $i$th option on trial $t$ depends on two inputs: 1) the RL module's current estimate of the option's value, $Q_{i,t}$, and 2) the relative gaze received by the option in the current trial. That is, we define a *linking function* [1] that maps learned Q-values and gaze onto the mean drift rates:

$$v_{i,t} = f(Q_{i,t}, \; gaze_{i,t})$$

(1)

where $gaze_{i,t}$ is defined as the proportional amount of time that option $i$ is fixated in trial $t$ during the time period from trial onset to choice (see Gaze effects). The linking function is important because it connects the RL and LBA modules together, and by changing the form of this function we can construct different variants of the model.

We tested eight different linking functions in the present study, which resulted in eight distinct models. Full descriptions of the models are provided in S1 Appendix. Below, we briefly summarize the two main dimensions on which they differ. First, some of the models assume a linear integration of Q-values into the mean drift rates (e.g., $v_{i,t} = \beta_Q \cdot Q_{i,t}$), while others assume a nonlinear integration. The models that assume a nonlinear integration use the softmax function to map Q-values onto the mean drift rates. Prior studies have demonstrated that a nonlinear mapping from Q-values to drift rates, and specifically the softmax, improves the fit of RL-SSMs [20,26,28,29]. The softmax causes the mean drift rates to depend on the *differences* between Q-values, similar to models that accumulate the relative evidence for choosing one option over another (e.g., the aDDM [14]; see S1 Appendix). An advantage of using the softmax, as opposed to raw Q-value differences, is that it enforces nonnegativity of the mean drift rates, which increases the probability that at least one accumulator will reach the decision threshold. With appropriate parameter settings, the softmax function can be very sensitive to small differences in the underlying Q-values, producing sharper decision boundaries compared to linear integration models (S1 Fig, left panels). The softmax also causes predicted response times to depend only on the difference between Q-values, with longer RTs for difficult choices (i.e., small Q-value differences). Linear integration models instead predict a magnitude effect: RTs are longer when both Q-values are small and faster when both Q-values are large (S1 Fig, right panels).

The second dimension on which the models differ is the effect of gaze on mean drift rates. In some of the models, relative gaze has an additive effect that is independent of learned option values: As an option receives a greater proportion of the total fixation time leading up to a choice, the likelihood that it will be chosen increases, regardless of its value. Other models assume a multiplicative effect in which gaze amplifies the value of the fixated option, similar to the aDDM [14]. Some of the multiplicative models predict a stronger effect of gaze on choice when the options have larger Q-values, but others do not predict this (S2 Fig). Although there is considerable evidence for a multiplicative gaze effect in the literature, the evidence from RL tasks is less clear [22]. One study that examined gaze effects in the test phase of an RL task (i.e., after learning had taken place) found support for an additive gaze effect [19]. In another RL study, a large subset of participants exhibited choice behavior that was inconsistent with core predictions of the aDDM [36], but an additive model was not explicitly considered. It is difficult to directly compare our model to the aDDM due to differences in the underlying mathematics (e.g., racing accumulators versus drift diffusion [37]). However, we were able to test several variants of a multiplicative gaze mechanism within a linear ballistic accumulator framework. As will be discussed below, we find mixed evidence for additive versus multiplicative gaze across our two experiments.

In addition to selecting actions, our models must be able to learn from the resulting outcomes. This is the responsibility of the RL module. After a choice is made and feedback is presented, the RL module updates the expected values for each option using a standard delta learning rule [38]:

$$Q_{i,t+1} = Q_{i,t} + \alpha \cdot \left( R(x_{i,t}) - Q_{i,t} \right)$$

(2)

where $Q_{i,t+1}$ is the updated expected value for the $i$th option, $Q_{i,t}$ is the previous expected value, and $\alpha$ is the learning rate parameter. The goal of this learning rule is to minimize the reward prediction error, $R(x_{i,t}) - Q_{i,t}$, or the difference between observed and expected outcomes for each option. Based on a growing body of research on context-dependent valuation in RL [30–32], our model assumes that the subjective value of $x_{i,t}$, the outcome from option $i$ on trial $t$, is a weighted combination of two range-normalized values:

$$R(x_{i,t}) = (1 - w_{rel}) \cdot \frac{x_{i,t} - min(x_{..})}{max(x_{..}) - min(x_{..})} + w_{rel} \cdot \frac{x_{i,t} - min(x_{.t})}{max(x_{.t}) - min(x_{.t})} \tag{3}$$

where the first term uses the minimum and maximum outcomes across all trials, and the second term uses the minimum and maximum outcomes on the current trial alone. Consider Option B in the task from Experiment 2 (Fig 1A). Option B's outcomes are low with respect to the global range of outcomes across all contexts; thus, it will produce lower values for the first range-normalized term. However, it produces larger outcomes than Option A most of the time, and because these two options are always presented together, Option B will frequently have high values for the second range-normalized term (i.e., 1.0). As the relative encoding parameter $w_{rel}$ approaches 1, the second term will dominate and options will be subjectively valued based on their *relative* rank within the local context. As $w_{rel}$ approaches 0, the first term will dominate and subjective values will reflect global ranks (context-independent). The transfer test in Experiment 2 is critical for estimating $w_{rel}$, as it pits options whose values were learned in one context against options whose values were learned in a different context. Because there was no transfer test in Experiment 1, for that experiment we fixed $w_{rel}$ to 0 (absolute encoding); however, the conclusions are the same if we instead fix $w_{rel}$ to 1 (relative encoding).

To summarize, our model includes up to eight free parameters: learning rate $\alpha$, relative encoding $w_{rel}$, drift rate scaling parameters $\beta_Q$ and $\beta_{gaze}$, softmax inverse temperature $\theta$, start point upper bound $A$, decision threshold $b$, and nondecision time $t_0$. The first two control the RL module and the last six control the LBA module. Not all models include the $\beta_{gaze}$ parameter, and only models with the softmax linking function include $\theta$. If $w_{rel}$ is nonidentifiable due to the task design, it can be set to 0.

## Model comparison

A model should not only fit the training data well, but should also generalize to unseen data from the same data-generating process [39]. With this in mind, we compared a set of candidate models using a metric known as accumulative one-step-ahead prediction error (APE) [40]. APE is a measure of out-of-sample prediction accuracy appropriate for time series data. It measures a model's ability to predict the next observation in a sequence, $y_{n+1}$, given only the previous observations, $y_1, ..., y_n$. Unlike similar metrics such as leave-one-out cross validation, APE only uses past observations to predict the future; it never uses future observations to predict the past. This makes it well-suited for RL problems. Further, because it evaluates predictions on unseen data, APE adjusts for model complexity in a way that depends on a model's functional form, not just the number of parameters. For a discussion of its other advantages and connections to other approaches, see [40].

Formally, the APE with logarithmic loss for model $M_j$ is defined as

$$APE(M_j) = \sum_{i=1}^{N-1} -\ln \hat{p}_{M_j}(y_{i+1}|y_1, ..., y_i) \tag{4}$$

where $\ln \hat{p}_{M_j}(y_{i+1}|y_1, ..., y_i)$ represents the log-likelihood that model $M_j$ assigns to observation $i + 1$, given the first $i$ observations. APE is straightforward to implement, albeit computationally expensive: For an individual data set with $N$ observations (i.e., choice-RT pairs), we fit the model $N - 1$ times, each time fitting a larger subset of the data. On each iteration,

we use the fitted parameters to compute the log-likelihood of the next unseen observation. Then, the one-step-ahead log-likelihoods are summed across observations. The model with the lowest APE is taken as the one that generalizes best to unseen data. In practice, it may not be possible to estimate the model parameters precisely until the model is fit to a minimum amount of data [40]. For the present application, we set the minimum number of data points to $n = 5$ (i.e., the sum in Eq 4 is from $i = 5$ to $N - 1$).

We used APE to compare the eight models described in S1 Appendix. The models use either a linear or nonlinear (softmax) linking function and assume either no gaze effect, a multiplicative gaze effect, or an additive gaze effect. As shown in Fig 2A, the best model in Experiment 1 was Model 7 ("softmax(Q) + gaze"), which had a significantly lower mean APE than each of the other models (one-sample t-tests on APE difference scores [$H_0 : \mu = 0$]; ps < .013). This model assumes a nonlinear mapping from Q-values to mean drift rates along with an independent, additive gaze effect. In Experiment 2, the best model was Model 8 ("softmax(Q + gaze)"), which had a significantly lower mean APE than each of the other models (ps < .0014). This model also assumes a nonlinear mapping from Q-values to mean drift rates. Although it may appear to be an additive gaze model, the softmax transformation causes the exponentiated Q values and gaze values to exert a

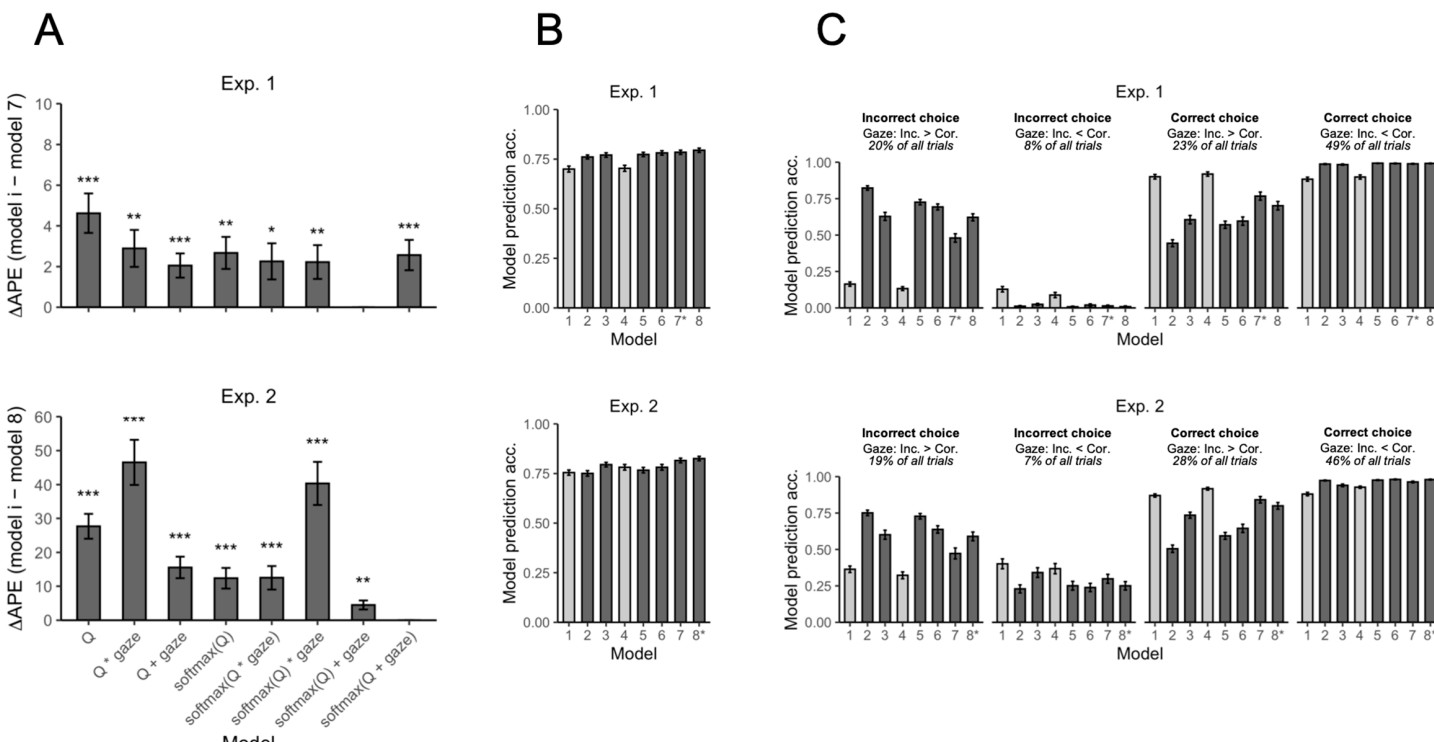

**Fig 2. Model comparison results. (A)** Each bar shows the mean difference in accumulative one-step-ahead prediction error (APE) between one of the models and the winning model, averaging across participants. The winning model, which had the lowest mean APE, differed between experiments (Exp. 1: "softmax(Q) + gaze"; Exp. 2: "softmax(Q + gaze)"). In all comparisons, the mean difference in APE was significantly above zero, indicating that the winning model generalized better to unseen data. *p < .05, **p < .01, ***p < .001 (one-sample *t*-tests). **(B)** Mean choice prediction accuracy for each model, averaged across participants. The models were simulated 100 times in the task using each participant's estimated parameters. On any given trial, a model's prediction was labeled correct if either of the following were true: (i) the participant made a correct (value-maximizing) choice and the model's proportion of correct choices on that trial across the 100 simulations was greater than 0.5, or (ii) the participant made an incorrect (nonmaximizing) choice and the model's proportion of correct choices was less than 0.5. **(C)** Mean choice prediction accuracy for each model as a function of whether the participant's choice was correct (maximizing) or incorrect (nonmaximizing), and whether the participant gazed longer at the correct or incorrect symbol before choosing. In all panels, error bars represent ±1 standard error of the mean. The darker bars represent models constrained by gaze data. In panels B and C, models are numbered in the same order as in panel **A**. The winning model is denoted with an asterisk.

multiplicative influence on mean drift rates (see S1 Appendix). Thus, softmax(Q + gaze) is actually a multiplicative gaze model. We conclude that while the evidence for a nonlinear relationship between Q-values and drift rates was consistent across experiments, the evidence for additive versus multiplicative gaze was mixed.

Fig 2B shows how accurate each model was at predicting choice, averaging across participants. Models constrained by gaze data (dark gray bars) predicted choice more accurately, on average, than models without gaze data (light gray bars). This was especially true in Experiment 1, where the gaze models achieved nearly 6–10% higher mean accuracy than the no-gaze models. Fig 2C provides more detail, breaking down model prediction accuracy into four types of trials based on whether the participant made a correct (maximizing) or incorrect (nonmaximizing) choice, and whether they gazed longer at the correct or incorrect symbol before choosing. Models constrained by gaze data predicted choice more accurately, on average, when the participant chose the symbol they looked at longer, which characterized about 65–70% of all trials. Models without gaze, in contrast, were more accurate when the participant made a correct choice, regardless of which symbol they looked at longer. The advantage of the gaze models is especially clear on trials where the participant looked longer at the incorrect symbol and subsequently selected it: Most of the gaze models predicted these erroneous choices better than chance (.50), whereas the no-gaze models were well below chance (see S3 Fig for an example of how one of the gaze models predicted an individual participant's choices on a trial-by-trial basis). Lastly, Fig 2C suggests that the multiplicative gaze models (Models 2, 5, 6, and 8) were more strongly affected by gaze than the additive gaze models (Models 3 and 7). In particular, multiplicative models were more accurate than additive models on trials where the participant looked longer at the incorrect symbol and subsequently chose it, but they were less accurate than additive models on trials where the participant looked longer at the incorrect symbol before choosing the correct symbol.

Fig 3A and Fig 3B show the distributions of the parameter estimates in both experiments from the softmax(Q) + gaze model and the softmax(Q + gaze) model, respectively. We make four observations. First, the distributions of the drift scaling parameters $\beta_Q$ and $\beta_{gaze}$ are mostly above zero, suggesting that both learned option values and proportional gaze influenced evidence accumulation. The means of these parameters were significantly greater than zero in both experiments and in both models (ps < .001; one-sample $t$ tests). Second, the smaller learning rates, higher decision thresholds, and higher nondecision times in Experiment 1 might reflect the greater complexity of the learning environment

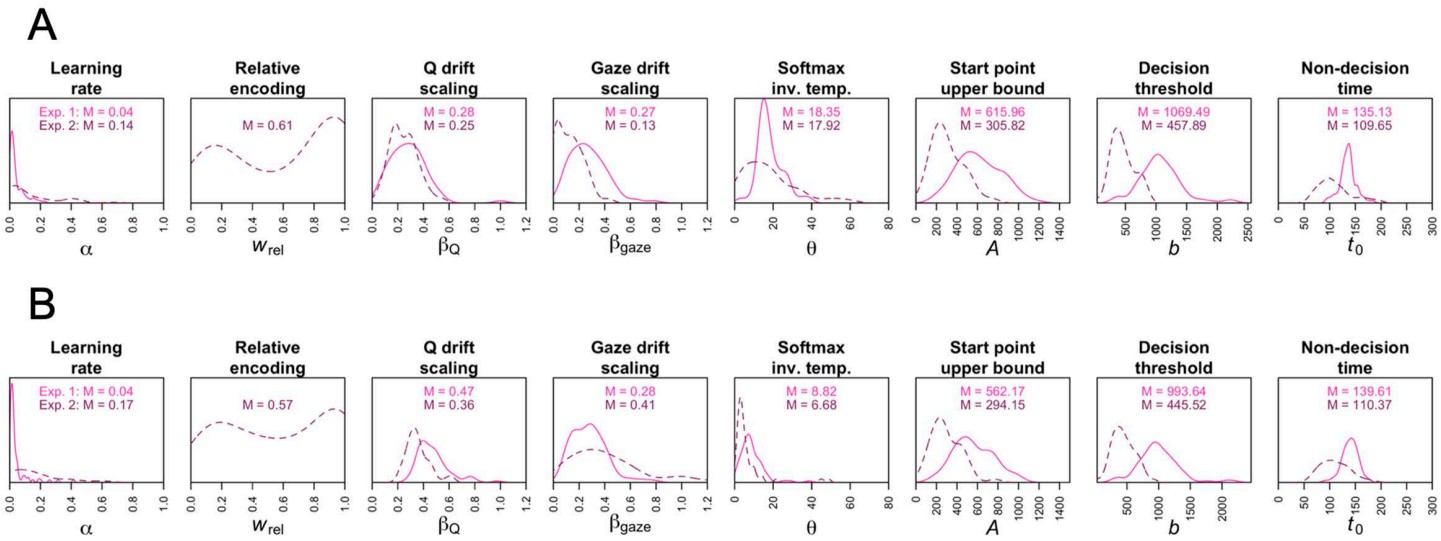

**Fig 3. Distributions of estimated model parameters.** Kernel density plots of the fitted parameters from (**A**) the softmax(Q) + gaze model and (**B**) the softmax(Q + gaze) model in both experiments. The relative encoding parameter was fixed to 0 in Experiment 1.

(Fig 1A). The higher thresholds and nondecision times are consistent with the slower mean RTs in that experiment (see next section). However, it should be noted that in Experiment 2, several nondecision time estimates were below biologically plausible minimum RTs for humans (<100 ms; [28]), and therefore should be interpreted with caution. Third, the average decision threshold was well above the average start point upper bound in both experiments. In the LBA model, this will lead to incorrect responses being slower than correct responses [13], which is consistent with our data: Incorrect choices were about 400–500 ms slower than correct choices, on average. Finally, the distribution of the relative encoding parameter was bimodal in the second experiment: Although the majority of participants exhibited relative outcome encoding (context dependence), there were some who formed more absolute-like representations. See the Supporting Information for results of parameter recovery simulations. In particular, we find that $\beta_Q$, $\beta_{gaze}$, and $w_{rel}$ are highly recoverable (true/recovered parameter correlations between 0.86 and 0.99; S4 Fig and S5 Fig).

**Choice-RT patterns**

Next, we examined the model's ability to fit basic choice-RT patterns in both experiments. In early trials, participants selected the correct, reward-maximizing symbols at near-chance levels. By the end of the learning phase, choice accuracy was well above chance. The best model for each experiment closely fit the learning curves (Fig 4A, 4E; purple lines). In contrast, the "Q + gaze" model, which has a linear linking function, underestimated choice accuracy, especially in later trials (green lines). This result held across models with different gaze mechanisms and in both experiments: "Linear" models consistently underestimated accuracy (S6–S9 Fig).

Choices also tended to become faster across the learning phase (Fig 4B, F). Although the model predicts this, the predicted effect was weaker than the empirical effect, especially in Experiment 2. In our model, choices become faster across trials because the mean drift rate for the correct option increases in magnitude as the underlying Q-values diverge. But faster RTs could also result from individuals reducing their response caution as they grow more accustomed to the task. In RL-SSMs, this can be implemented by allowing the decision threshold to decrease across trials [25]. Indeed, a modified version of our model with a trial-dependent, decreasing decision threshold fit the aggregate RT curve better (see S2 Appendix and S7 Fig). Thus, the assumption of a static decision threshold may be too restrictive if the goal is to closely fit RT curves. Since this was not a theoretical priority in the current study, we chose not to formally evaluate the decreasing threshold models with APE. Even with a static threshold, however, the winning models provided a good fit to the aggregate RT distributions for correct (maximizing) and incorrect (nonmaximizing) choices (S10 and S11 Figs).

Our model also accounts for heterogeneity across individuals. The winning model explained 87% of the variance in individual choice accuracies and 99% of the variance in individual mean RTs in the learning phase of Experiment 1 (Fig 4C, 4D). The percentages were 84% and 86%, respectively, in the learning phase of Experiment 2 (Fig 4G, 4H). Several model parameters were significant predictors of choice accuracy during learning. In Experiment 1 (best model: softmax(Q) + gaze), higher choice accuracy was associated with higher learning rates ($a$), higher Q-value drift coefficients ($\beta_Q$), lower gaze drift coefficients ($\beta_{gaze}$), higher softmax inverse temperatures ($\theta$), and higher decision thresholds ($b$) (ps < .039; see multiple regression results in S1 Table). In Experiment 2 (best model: softmax(Q + gaze)), higher learning phase accuracy was predicted by higher learning rates, stronger relative encoding ($w_{rel}$), lower start point upper bounds, and higher thresholds (ps < .018; S2 Table). Several model parameters were also related to response times during learning: Longer mean RTs were associated with lower values of $\beta_Q$, lower values of $\beta_{gaze}$, lower start point upper bounds, and higher decision thresholds in Experiment 1 (ps < .001; S3 Table); and with lower $\beta_Q$, higher decision thresholds, and longer nondecision times ($t_0$) in Experiment 2 (ps < .008; S4 Table). Differences between experiments may be due in part to the slightly different functional roles that specific parameters play in the softmax(Q) + gaze and softmax(Q + gaze) models (e.g., $\beta_{gaze}$; see S1 Appendix).

Next we turn to the transfer test in Experiment 2, which was designed to reveal context-dependent valuation in RL. Recall that in the learning phase, each symbol was either the "better" option in its original context or the "worse" option.

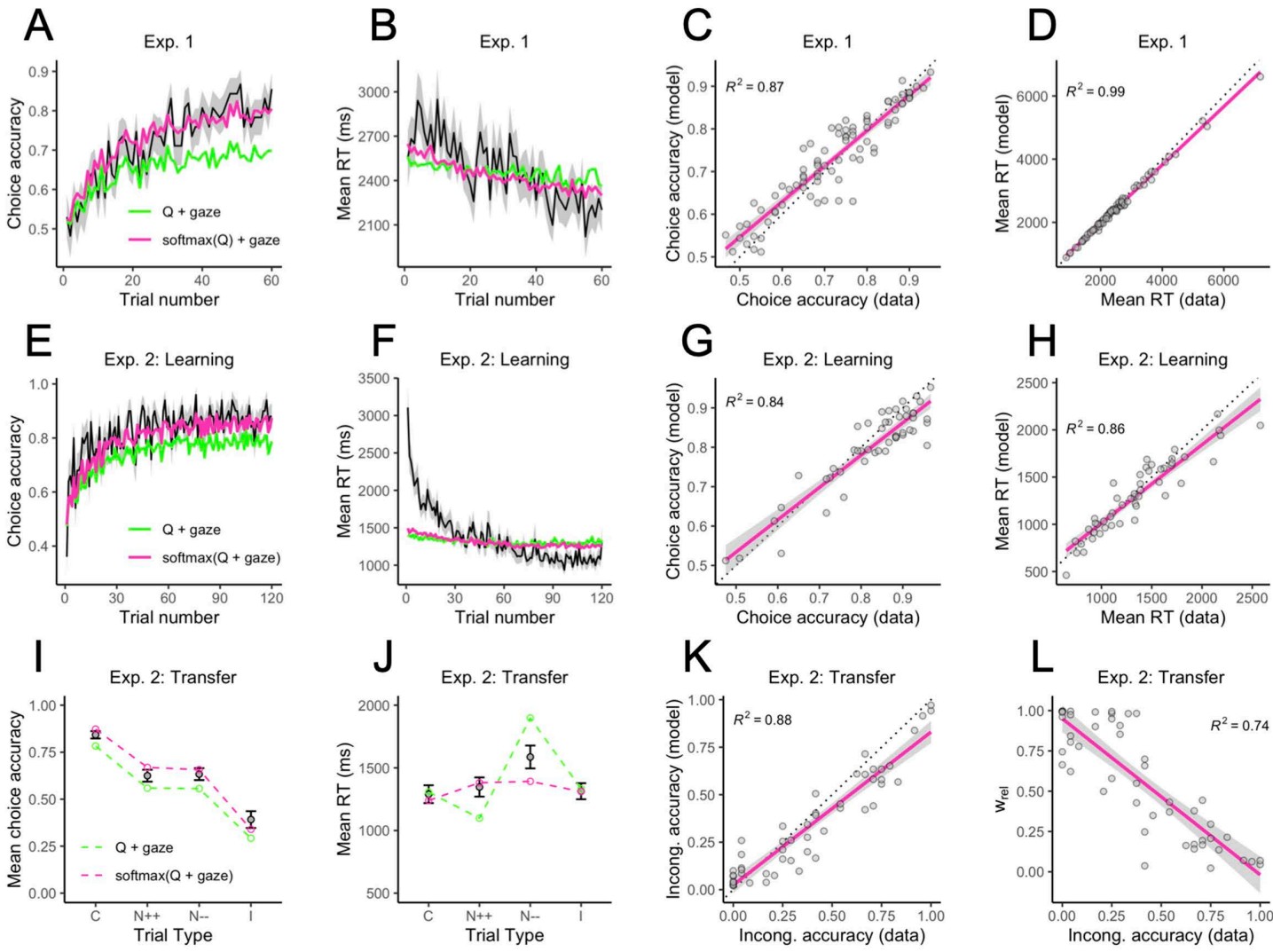

**Fig 4. Choice-RT effects. (A–D)** Proportion of correct choices on each trial, mean RT on each trial, and relationships between empirical and fitted individual choice accuracies and mean RTs in Experiment 1. **(E–H)** The same as panels A–D, but for the learning phase in Experiment 2. **(I–J)** Mean choice accuracy and mean RT for the four types of trials in the transfer test of Experiment 2 (C = congruent, N++ = neutral with both options having high relative values, N-- = neutral with both options having low relative values, I = incongruent). **(K)** Relationship between empirical and fitted individual choice accuracies on incongruent trials. **(L)** Relationship between incongruent choice accuracies and the estimated relative encoding parameter, $w_{rel}$. In all panels, error ribbons and error bars represent ±1 standard error.

The transfer test involved choices between all possible pairwise combinations of symbols. Thus, the transfer choices can be categorized into four types: A higher-valued "better" option paired with a lower-valued "worse" option (Congruent trials), two "better" options (Neutral++ trials), two "worse" options (Neutral-- trials), or a higher-valued "worse" option paired with a lower-valued "better" option (Incongruent trials). Our model successfully captured the characteristic pattern of context-dependent valuation: High choice accuracy on Congruent trials, low accuracy on Incongruent trials, and intermediate accuracy on Neutral trials ([Fig 4I]). The winning model provided a close fit to the mean RTs in three of the four categories; however, it underestimated the longer RTs on Neutral-- trials ([Fig 4J]; purple line). Interestingly, the "Q + gaze" model (green line) underestimated RTs on Neutral++ trials, where both options have higher Q-values and drift rates, and

overestimated RTs on Neutral-- trials, where both options have lower Q-values and drift rates (that is, assuming $w_{rel} \gg 0$). We would expect the aDDM, which predicts faster decisions for higher overall value [22], to also produce faster RTs on Neutral++ trials and slower RTs on Neutral-- trials. Our nonlinear models do not exhibit this behavior because the softmax removes any distinction between two high Q-values and two low Q-values: only the difference matters. Although none of our models perfectly captured the pattern of mean RTs across the transfer choice categories, this is one aspect of the data that the linear integration models seem to capture better than the softmax models (S12 Fig).

Lastly, there was considerable individual variability on Incongruent trials: Some participants were highly accurate in choosing the correct option, but many performed well below chance. The softmax(Q + gaze) model explained 88% of the variability across individuals (Fig 4K). As expected, $w_{rel}$ was the strongest predictor of Incongruent trial accuracy out of the model parameters, with higher $w_{rel}$ associated with lower accuracy (Fig 4L; see multiple regression results in S5 Table).

### Gaze effects

Gaze effects were examined as follows. First, we computed, for each participant and each trial, the proportional amount of time that each symbol was fixated during the time period from trial onset to choice. For example, if on a particular trial there were three fixations on the left symbol with durations of 260, 136, and 180 ms, and one fixation on the right symbol with a duration of 224 ms, the proportional gaze for the left symbol would be $(260 + 136 + 180)/(260 + 136 + 180 + 224) = .72$ and the proportional gaze for the right symbol would be $224/(260 + 136 + 180 + 224) = .28$. Nonsymbol fixations and fixations on unavailable symbols (Experiment 1) were excluded from the denominator. If neither of the available symbols was fixated, both received a proportional gaze score of 0. We then computed the difference between the proportional gaze scores for the correct (higher valued) and incorrect (lower valued) symbols on each trial. This difference score ranges from 1 to −1, with positive values indicating a relative gaze advantage for the correct symbol, and negative values indicating a relative gaze advantage for the incorrect symbol.

Previous studies have shown that relative gaze predicts choice: the longer an item is fixated, the more likely it is to be chosen [14,15,18–20]. Based on this, we should expect higher choice accuracy on learning phase trials where the correct option received relatively more gaze than the incorrect option. We grouped the gaze difference scores defined above into five quintiles, separately for each participant, and plotted the mean choice accuracy and mean RT in each quintile. Choices in the learning phase were not only more accurate, but also faster when the correct option received a greater proportion of the total fixation time, consistent with a drift rate effect (Fig 5A, 5B, 5E, 5F). Our best gaze-augmented models (purple lines) captured these effects much better than the models without gaze, such as the "softmax(Q)" model (blue lines). See S13–S15 Figs for the fits of all models to the gaze effects in both experiments. Our best models were also able to explain 66% (Exp. 1) and 78% (Exp. 2) of the variability across individuals in the magnitude of the gaze effect on choice, measured as the difference in choice accuracy between the 5th gaze quintile and the 1st gaze quintile (Fig 5C, 5G). Individuals with higher values of the $\beta_{gaze}$ parameter—or the product $\theta\beta_{gaze}$ for the "softmax(Q + gaze)" model (see S1 Appendix)—tended to show larger gaze effects on choice (Fig 5D, 5H).

On average, the correct option had a significant proportional gaze advantage (estimated intercept = 0.059 [Exp. 1] and 0.20 [Exp. 2], ps < .001) that increased across the learning phase (trial number: slope = 0.018 [Exp. 1] and 0.058 [Exp. 2], ps < .001) and was greater for choices involving options with larger value differences (EV difference: slope = 0.035, p < .001 [Exp. 1]; linear mixed-effects regressions on proportional gaze difference scores; S6 Table and S7 Table). Thus, there is a potential confound: When the subjective values of the available options are more distinct, either because the decision maker has had more experience with the task (a learning effect) or because the underlying reward distributions are spaced far apart (a difficulty effect), the better option will not only be selected faster and with greater probability, but may also receive greater attention precisely because it is much more valuable. The important question for us is whether gaze affects choice after controlling for learning and difficulty effects.

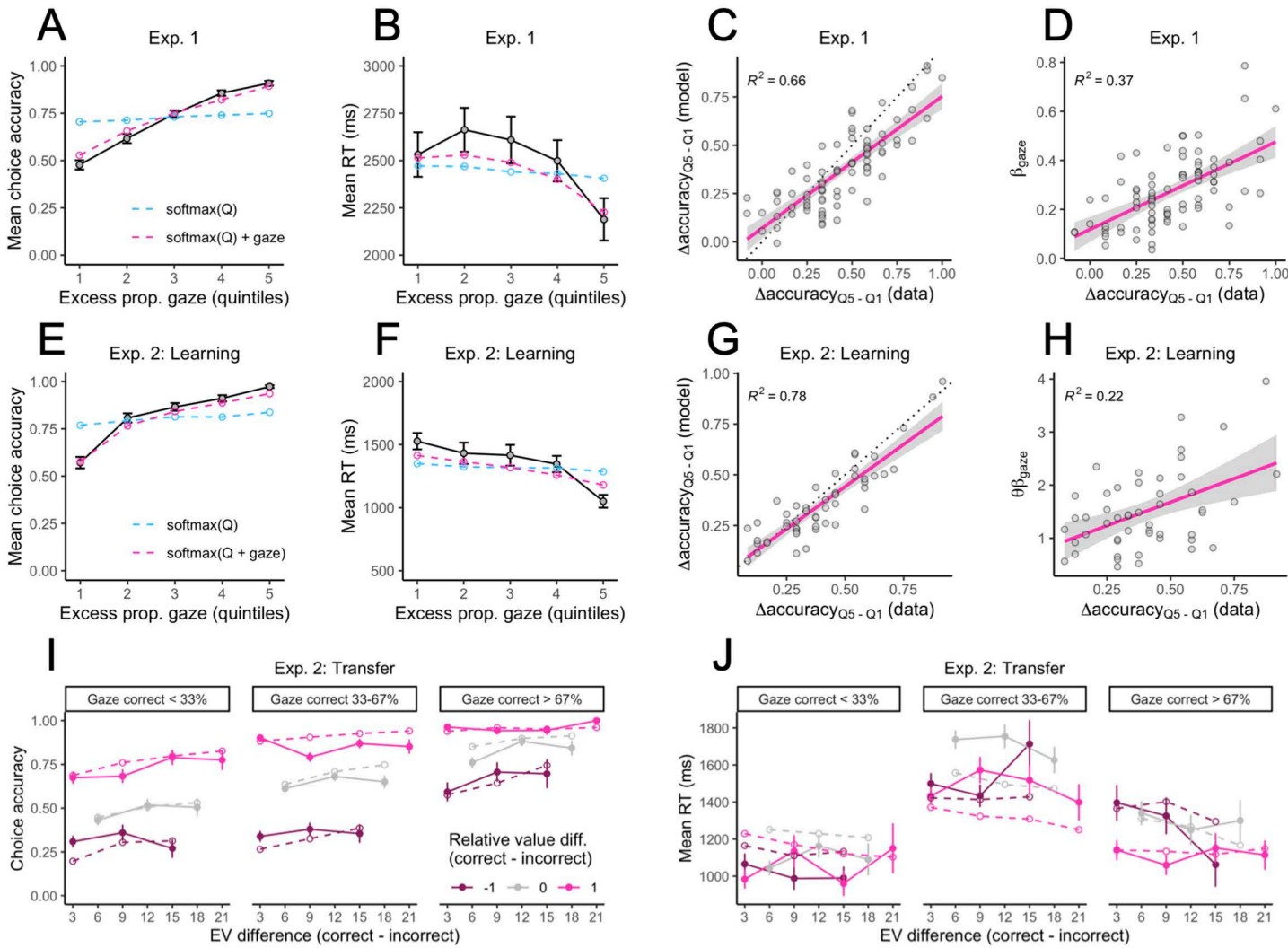

**Fig 5. Gaze effects. (A–B)** Mean choice accuracy and mean RT as a function of the excess proportional gaze on the correct option in Experiment 1, broken up into five quintiles. The quintiles were constructed by taking the difference between the proportional gaze on the correct (higher valued) and incorrect (lower valued) symbols on each trial, sorting the difference scores, and dividing them into five equal-sized bins, separately for each participant. The higher the quintile, the longer the correct option was fixated relative to the incorrect option. **(C)** Relationship between empirical and fitted gaze effects at the individual level in Experiment 1, defined as the difference in accuracy between the most extreme gaze quintiles. **(D)** Relationship between individual gaze effects and the gaze drift scaling parameter, $\beta_{gaze}$, from the "softmax(Q) + gaze" model in Experiment 1. **(E–H)** Experiment 2 gaze effects in the learning phase (same as panels A–D). In panel H, individual gaze effects are plotted against the product of the softmax inverse temperature and the gaze parameter from the "softmax(Q + gaze)" model: $\theta\beta_{gaze}$. **(I–J)** Proportion of correct choices and mean RT as a function of the expected value difference between the available options, the relative value difference, and the proportional gaze on the correct, higher-valued option in the transfer test of Experiment 2. Dashed lines show model fit.

To test this, we used logistic mixed-effects regression to model the probability of choosing the correct symbol on each learning phase trial as a function of trial number, the EV difference between the available options (correct minus incorrect; Experiment 1 only), the overall EV between the available options (correct + incorrect) and the proportional gaze difference for the available options (correct minus incorrect). The two-way interactions with trial number were also included. EV difference was not included in the model for Experiment 2 because it was constant across trials (ΔEV = 3). In both

experiments, the proportional gaze difference was a significant predictor of choice accuracy even after controlling for the other predictors (gaze difference: slope = 1.076 [Exp. 1] and 1.33 [Exp. 2], ps < .001). The gaze effect weakened across trials in Experiment 1 (trial number × gaze difference: slope = -0.14, p = 0.022), but not in Experiment 2 (slope = -0.058, p = 0.41). Taken together, these results indicate that relative gaze influenced learning phase choice accuracy over and above the effects of trial number, the separation between option values, and the sum of option values.

A parallel linear mixed-effects model was performed with log-transformed response time (log-RT) as the dependent variable. In both experiments, greater gaze toward the correct option predicted faster responses even after accounting for the other predictors (gaze difference: slope = −0.031 [Exp. 1] and −0.071 [Exp. 2], ps < .001). The negative effect of gaze on log-RT strengthened across trials in Experiment 1 (trial number × gaze difference: b = −0.0097, p = 0.065), but weakened across trials in Experiment 2 (slope = 0.027, p = 0.0018). The key takeaway is that, as with choice accuracy, relative gaze influenced response times in the learning phase over and above the effects of trial number, EV differences, and overall EV.

To analyze gaze biases in the transfer test of Experiment 2, we used logistic mixed-effects regression to model the probability of choosing the correct, maximizing symbol on each trial as a function of the difference in expected values, the difference in relative values, the overall (summed) expected value, the overall relative value, and the difference in proportional gaze between the available symbols (correct minus incorrect). For this analysis, relative value was based on a symbol's rank within its original learning context (0 = worse option, 1 = better option). The relative value difference can then take values of +1 (Congruent trials), 0 (Neutral trials), or -1 (Incongruent trials).

Relative value differences had a large impact on choice accuracy in the transfer test (slope = 1.20, p < .001), more so than expected value differences (slope = 0.27, p = .010). There was also a robust gaze effect: Holding values constant, accuracy increased with the proportional gaze allocated to the correct option (Fig 5I). In the mixed-effects regression, proportional gaze difference was a significant predictor of choice accuracy over and above the other predictors (slope = 0.89, p < .001). The softmax(Q + gaze) model closely reproduced the combined effects of absolute expected values, relative values, and gaze asymmetries on transfer choice accuracy at the group level (Fig 5I).

Mean RTs in the transfer test were noisier and more difficult for our model to fit. Visual inspection of Fig 5J suggests two major trends: 1) Slower RTs on Neutral trials (relative value difference = 0) compared to Congruent and Incongruent trials (relative value difference = $\pm$1), and 2) slower RTs on trials with a more even gaze allocation between the two symbols. Thus, we analyzed log-RTs using linear mixed-effects regression with the same predictors as the model for choice accuracy, except that the *unsigned* relative value difference and *unsigned* proportional gaze difference were used instead of the signed versions. Unsigned proportional gaze difference was a significant negative predictor of log-RT, controlling for the other predictors (slope = -0.08, p < .001). Unsigned relative value difference (slope = -0.05, p < .001) and overall relative value (slope = -0.05, p < .001) were also significant negative predictors of log-RT. Thus, RTs were slower whenever relative values or gaze proportions were (approximately) equal. Our computational model qualitatively captures these patterns. However, as previously discussed, the model struggled to predict the negative effect of overall relative value (i.e., slower RTs when both options have low relative value).

## Discussion

In this study, we introduced a reinforcement learning and sequential sampling model (RL-SSM; [1]) that can account for multiple behavioral regularities in decisions from experience. In addition to capturing choice-RT patterns and learning effects, our model uses exogenous gaze data—the proportional fixation time for each option—to constrain the evidence accumulation process on each choice trial [14,15,19,20]. This allows it to account for gaze effects that would be missed by existing RL-SSMs [25–29], including the tendency for people to choose the options that they look at more [14,15,18–21]. Although this effect has been documented in RL settings [9,19], this study is the first, to our knowledge, to incorporate gaze biases into a full RL-SSM that simultaneously captures learning and choice-RT effects.

Constraining our model with gaze data enables it to make better predictions. In particular, it correctly predicts that reward-maximizing options are more likely to be selected when they receive relatively more gaze prior to choice (Fig 5A, 5E). Further, there were occasional trials in which participants would fixate longer on the nonmaximizing option before selecting it. Models that do not incorporate gaze data struggled to predict these erroneous choices; however, our gaze-augmented models correctly predicted many of them due to the influence of relative gaze on mean drift rates (Fig 2C). Our model also predicts gaze effects on choice RT: Choices were generally faster when the higher-value option received a greater share of the total fixation time (Fig 5B, 5F). This effect is naturally explained by the influence of gaze on mean drift rates. Thus, the results of this study demonstrate the clear benefits of incorporating eye gaze data into models of experience-based decision making [8,9].

Central to our model is the linking function that maps learned Q-values and relative gaze onto mean drift rates. We tested a number of different linking functions that differ on two key dimensions: First, whether Q-values are mapped linearly or nonlinearly onto mean drift rates, and second, whether gaze has no effect, an additive (independent) effect, or a multiplicative effect on mean drift rates. Previous studies have found support for a nonlinear mapping between option values and drift rates in sequential sampling models [20,26,28,29]. Nonlinearity seems to be important for capturing choice accuracy rates, as linear versions of our model consistently underestimated accuracy. The softmax function that our models use to create this nonlinear mapping causes the mean drift rate for a given option to vary inversely with the learned values of the other available options, thereby inducing competition between the racing accumulators (see S1 Appendix). As noted by [28], the softmax could then be interpreted as a kind of lateral inhibition between the available choice options. Lateral inhibition is a biologically inspired mechanism that allows models with separate racing accumulators to emulate diffusion models, which accumulate the *relative* evidence for choosing one alternative over the other [12]. Thus, we would expect our model to behave similarly to diffusion models despite using separate racing accumulators.

Note that the relativization of mean drift rates via the softmax is separate from the relativization of outcomes in the RL module, which is entirely controlled by the $w_{rel}$ parameter (Eq 3). The former operates at the decision stage and ensures that mean drift rates depend on the relative differences among Q-values, regardless of what the Q-values represent. The latter operates at the outcome encoding stage and allows the model to account for a specific value learning bias; that is, the tendency for learned Q-values to reflect context-dependent ranks, rather than absolute expected values [31,41,42]. The combination of these separate relativization mechanisms enables the model to produce and fit a wide range of behavioral effects.

Our model comparisons favored an additive gaze mechanism over a multiplicative gaze mechanism in the first experiment, consistent with some prior work [19,21]. However, a multiplicative gaze mechanism was favored in the second experiment, which aligns with well-established models like the aDDM [14,22]. Psychologically, an additive gaze mechanism might reflect the influence of factors such as physical salience [21,43], estimation uncertainty [9], spatial location [44], or even random noise, all of which can bias choice independently of value. A multiplicative gaze mechanism might instead suggest that directly fixating on an option facilitates the computation of its value, or the integration of past outcomes sampled from memory [45,46]. Such a facilitation effect would tend to benefit high-value options more than low-value options, as is generally assumed by multiplicative gaze models [22]. However, the best-performing multiplicative gaze model in Experiment 2, softmax(Q+gaze), does not predict a stronger gaze effect for high value options (S2 Fig), which suggests that magnitude sensitivity [47] was not the reason for the model's advantage in this context. It is difficult to know why the evidence for additive versus multiplicative gaze was mixed across our two experiments. However, it should be noted that the two winning models provided very similar fits to the data in both experiments, and the advantage of one over the other appears to be quantitative rather than qualitative. Future studies should consider different experimental designs for which our additive and multiplicative gaze models would make qualitatively distinct predictions.

There are a number of important caveats in this discussion of gaze mechanisms. First, we only tested racing accumulator models in the present study, but additive and multiplicative gaze in a racing accumulator model may not function exactly like additive and multiplicative gaze in a diffusion model like the aDDM [14]. Future research should compare

additive and multiplicative gaze mechanisms across both accumulator and diffusion-based RL-SSMs using, for example, methods found in [37]. Second, additive versus multiplicative gaze is likely a false dichotomy. It could be that *both* mechanisms play a role in decision making. One study suggests that the mechanism of gaze may even change over the time course of decision formation as information accumulates toward a decision threshold. Early in a decision, gaze can amplify the influence of value information (a multiplicative effect), but later, gaze may simply correlate with the emerging choice (an additive effect) [48]. Thus, models combining both multiplicative and additive mechanisms may be useful to consider in future work.

Group-level model comparisons can sometimes obscure heterogeneity that exists between individuals. One way to approach individual differences in a model-based framework is to correlate the relative performance of one model over another with behavioral measures that should distinguish between those models (S3 Appendix). For example, although models with nonlinear linking functions were favored on average, some participants were better described by models with linear linking functions. Additional analysis indicated that these participants exhibited different choice-RT effects. For example, participants in the first experiment whose response times were faster when choosing between high-value options tended to be better described by a model with a linear integration of Q-values into mean drift rates ("Q + gaze"), as this model predicts a magnitude effect on RTs (S1 Fig). As another example, participants who exhibited a stronger effect of gaze when choosing between high-value options tended to be better described by a multiplicative gaze model that predicts such an interaction. Both of these effects have been documented in the literature [22,47,49]; however, they were not prominent enough in our data to give models with linear linking functions or magnitude-dependent gaze effects an advantage at the group level. Future work should consider ways of combining alternative mechanisms in the same model, with parameters to arbitrate between them. For example, it may be possible to construct a linking function that interpolates between the linear and softmax mappings, so that the same model could produce and fit a wider range of behaviors.

Our model has a number of practical advantages. First, because it uses linear ballistic accumulation [13], it has an analytically tractable likelihood function and can be easily fit to individual choice-RT data. As there is substantial variability between individuals in the degree to which gaze influences choice [20], we believe the ease of fitting our model to individual data is a major benefit. Second, unlike drift diffusion models, racing accumulator models can be readily generalized to choices between three or more alternatives without any additional assumptions. Third, we showed that most of the model's parameters are recoverable, even when fit to relatively small datasets (e.g., N = 60 observations; S4 Fig).

The present study also has important limitations. First, similar to other approaches [14,15,19,20], our model does not explain gaze behavior; instead, it treats gaze as an exogenous input, requiring the use of an eye-tracker to measure visual fixations. The model could be extended to also predict the relative gaze that each option receives on each trial. A good starting point would be to assume that relative gaze reflects (learned) option values [50]—which our data support—but a more complete account would incorporate other factors such as physical salience [21,43] or estimation uncertainty [9]. It is also possible that gaze may play a role in value updating, as attention during the feedback phase could influence subsequent value representations [8]. Second, because our model lumps fixations together in the calculation of relative gaze, it cannot account for fixation order effects, such as the tendency to choose the last-fixated option [14]. The piecewise linear ballistic accumulator framework [51] may provide a solution that would enable the model to make use of the full sequence of fixations on each trial. Finally, we have evaluated our model on data from two-alternative forced choice tasks with complete feedback. We believe that increasing the number of choice alternatives and varying the nature of outcome feedback in future experiments will provide even stronger tests of the model's core assumptions.

## Materials and methods

### Ethics statement

Written informed consent was obtained from all participants prior to the experiment. The study procedures were approved by the Institutional Review Board at Binghamton University.

## Participants

Undergraduate students participated in two experiments in exchange for partial course credit. In the first experiment, 83 students (52 women, 29 men, 2 transgender or another gender identity; ages 18–22, M = 19.05) completed a 20-minute eye-tracking task. In the second experiment, 50 students (39 women, 11 men; ages 18–27, M = 19.1) completed a longer eye-tracking task in approximately 45 minutes. Sample sizes for both experiments were determined through power analyses designed to detect a medium-sized within-subject effect (d = 0.50) with 90% power at an alpha level of .05. Participants were informed that points earned during the learning phase (Experiment 1) or the transfer test (Experiment 2) would be converted into candy, with the conversion rate explained in the instructions. All participants were at least 18 years old, reported normal or corrected-to-normal vision, and provided written informed consent prior to participating.

## Procedures

**Experiment 1.** Participants were instructed to make repeated choices between symbols with the goal of earning as many points as possible. The task was adapted from Experiment 1 in [31] and involved a learning phase with eight choice options grouped into two contexts of four options each. These contexts featured either a positively skewed or negatively skewed distribution of outcomes, with identical outcome ranges. Two identical options, each with an expected value (EV) of 30 points, were embedded in both contexts. The contexts were designed in this way to test competing theories of context-dependent valuation in RL (results not presented here). With four options per context, there were 12 unique option pairs across the two contexts. Each pair was presented five times, resulting in 60 total learning trials.

Context presentations were interleaved randomly across the learning phase. Symbols within a context shared a color (orange or blue) but differed in shape. The color-to-context and symbol-to-option mappings were randomized across participants.

Before beginning the learning phase, participants completed five practice trials with different symbols to familiarize themselves with the setup. Each learning trial began with a drift check to align the participant's gaze at the center of the screen. Then, the four symbols from one of the contexts were presented on screen, but only two symbols were available for selection (indicated by asterisks above the symbols). This was done to encourage learning about all options within a context rather than simply identifying the option with the highest expected value. The four symbols, each measuring 200 × 200 pixels, were positioned 270 pixels from the screen center in one of four directions: up, down, left, or right. This corresponded to a visual angle of 8.52 between the horizontally aligned symbols and 8.58 between the vertically aligned symbols. Participants made their choice using the arrow key corresponding to the location of their preferred symbol. If a participant attempted to select one of the unavailable symbols, the trial would not advance until a valid choice was made. After a valid choice, participants returned their gaze to center for 300 ms before seeing outcomes for all four symbols (full feedback). Outcomes were overlaid on semi-transparent symbols, with the selected symbol highlighted by a white box. Feedback viewing was self-paced, and participants advanced to the next trial by pressing the space bar. Outcomes were pre-generated for each option from Gaussian distributions centered around their respective EVs with a standard deviation of 2 (rounded to the nearest integer). Although points from the chosen options were added to a running total, the cumulative score was not displayed.

At the end of the learning phase, participants were shown their total accumulated points and the maximum/minimum possible points they could have earned by choosing the correct/incorrect option on every trial. The learning phase was followed by a memory-based value estimation task (results not presented here).

**Experiment 2.** In this experiment, participants learned fixed value differences within four pairs of symbols. One symbol in each pair had a consistently higher EV than the other. Across 120 randomized learning trials, each symbol was presented 30 times within its fixed pair. On each trial, participants chose between the two symbols from a randomly selected context pair and received complete feedback, with the outcomes for both the chosen and unchosen options

displayed just below each symbol. The two symbols, each measuring 300 × 300 pixels, were positioned 480 pixels to the left or right of the screen center. This corresponded to a visual angle of 15.1 between the horizontally aligned symbols. Participants were informed at the start of the learning phase to learn which symbols were most valuable. Participants made their choice between the pair by using the left or right arrow key. After each selection, participants returned their gaze to a center fixation for 300 ms before the points for both symbols were presented.

Participants then completed a transfer test that included choices between all 28 unique pairwise combinations of the eight symbols. No outcome feedback was presented during the transfer test to assess participants' learned value representations. Each unique pair was presented four times for a total of 112 transfer trials. Across both phases, symbols were counterbalanced such that each appeared an equal number of times on the left and right sides of the screen. During the transfer test, participants were instructed to choose the symbol they believed would yield more points on each trial. They were also informed that the points accumulated in this phase would be converted into candy, with the specific conversion rate displayed to indicate how many candies they could earn based on their accuracy in the test. Each transfer trial began with a drift check fixation, followed by two symbols presented side by side. Using the same arrow keys as in the learning phase, participants made their selection, and the chosen symbol remained highlighted for 0.5 seconds by a white box before the next trial began.

## Eye tracking

Fixation data was collected with SR Research EyeLink software (version 5.15). Gaze position was recorded from the right eye using a table-mounted EyeLink 1000 Plus eye tracker with the sampling rate set to 500 Hz. Stimuli were presented on a 24-inch HD LCD monitor with a 1920 × 1080 pixel resolution against a constant gray background. Participants sat approximately 100 cm from the screen with their heads stabilized in a chinrest under consistent, slightly dimmed lighting. In Experiment 1, a 9-point calibration and validation procedure was performed once at the beginning of the session. In Experiment 2, calibration and validation were performed at the beginning of the experiment and again before the start of the transfer test. Fixations were automatically identified using the EyeLink software, and rectangular areas of interest (AOIs) were defined around each symbol that measured 270 × 270 pixels in Experiment 1 and 340 × 340 pixels in Experiment 2. The average validation error was approximately 0.40 of visual angle in both experiments.

## Statistical analysis

We analyzed gaze effects using a series of mixed-effects regression models from the "afex" package in R [52]. Choice accuracy was analyzed with mixed-effects logistic regression, and proportional gaze difference and log-transformed RTs were analyzed with linear mixed-effects regression. All predictors were z-scored within-subject to facilitate interpretability and model convergence. We utilized a maximal random effects structure by including random intercepts and slopes for each within-subject predictor to account for heterogeneity across participants [53]. Almost all models successfully converged under this structure. However, one model did not converge, so we removed the random effect correlations. The conclusions were not affected by whether the random effect correlations were included. Estimated coefficients and statistical tests for all mixed-effects models are provided in the Supplemental Information (S6–S13 Tables).

## Computational modeling

RL-SSMs were fit to individual choice-RT data using maximum a posteriori (MAP) estimation [54] (in Experiment 2, the models were fit to the learning and transfer data). MAP estimation is very similar to maximum likelihood estimation, except it uses priors to regularize the model parameters. We chose independent, moderately informative priors based on past research (e.g., realistic values for nondecision time; [55]) and simulations of the model across its parameter space:

$$\alpha \sim Beta(1.3, 3.7)$$

$$w_{rel} \sim Beta(1.1, 1.1)$$

$$\beta_Q \sim Gamma(2, 0.5)$$

$$\beta_{gaze} \sim Gamma(2, 0.5)$$

$$\theta \sim Gamma(2, 20)$$

$$A \sim Gamma(6, 100)$$

$$b_{sep} \sim Gamma(6, 100)$$

$$t_0 \sim Gamma(6, 30)$$

where the gamma distributions have the shape-scale parameterization. Assigning a nonnegative prior to the threshold separation parameter $b_{sep}$ ensures that the start point upper bound ($A$) will not exceed the decision threshold ($b$), since $b = A + b_{sep}$. For all fits, we fixed the drift rate standard deviation $s$ to 0.1 and initialized Q-values to 0.5.

After assigning priors to the parameters, the MAP objective function becomes the sum of the log-likelihood and the log-prior (i.e., the log-posterior). The log-likelihood function for our RL-SSM is the same as that for the regular LBA model (see [13]), but with mean drift rates computed using the appropriate linking function (S1 Appendix). Log-likelihoods were summed across trials, skipping trials with RTs faster than 250 ms or slower than 10 s. We searched for parameters that optimized the MAP objective function using differential evolution optimization (NP = 100, itermax = 1000, steptol = 250) [56].

When simulating the RL-SSM, we used the current Q-values and proportional gaze scores for the available options to compute the mean drift rates $v_{i,t}$ on each trial according to the model's linking function. Start points $z_{i,t}$ for each accumulator were drawn independently from a $\mathcal{U}(0, A)$ distribution, and drift rates $d_{i,t}$ were drawn independently from $\mathcal{N}(v_{i,t}, s)$ distributions (resetting negative drift rates to zero). The time taken for the $i$th accumulator to reach the threshold can then be computed as

$$time_{i,t} = (b - z_{i,t})/d_{i,t}$$

i.e., time = distance / speed. The simulated choice is given by the index $i$ with the minimum $time_{i,t}$, and the simulated RT is given by $t_0 + \min_i(time_{i,t})$. This procedure was repeated until the simulated RT was at least 250 ms but no longer than 10 s. In the figures, the model predictions were averaged across 100 simulations using each participant's best-fitting parameters.

## Supporting information

**S1 Appendix. Model descriptions.**
(PDF)

**S2 Appendix. Trial-dependent decision threshold.**
(PDF)

**S3 Appendix. Model-based analysis of individual differences.**
(PDF)

**S1 Fig. Simulated choice-RT patterns in a two-alternative choice task.** Each model was simulated 500 times for each combination of Q-values (proportional gaze was set to 0.5 for both alternatives). The left panels show the proportion of times the second alternative was chosen (prop2) and the right panels show the mean RT (ms) across the 500 runs for each Q-value combination. Models were simulated using the mean parameter estimates from Experiment 1 (Q+gaze: $\beta_Q$ = 0.32, $\beta_{gaze}$ = 0.27, $A$ = 631.09, $b$ = 1086.19, $t_0$ = 133.44; softmax(Q) + gaze: $\beta_Q$ = 0.28, $\beta_{gaze}$ = 0.27, $\theta$ = 18.35, $A$ = 615.96, $b$ = 1069.49, $t_0$ = 135.13; softmax(Q+gaze): $\beta_Q$ = 0.47, $\beta_{gaze}$ = 0.28, $\theta$ = 8.82, $A$ = 562.17, $b$ = 993.64, $t_0$ = 139.61). The Q+gaze model exhibits a noisier decision boundary and produces mean RTs that depend on the magnitude of the Q-values, with faster RTs for larger magnitudes. The softmax models exhibit a sharper decision boundary and mean RTs that depend only on the difference between Q values, with faster RTs for larger differences.
(PNG)

**S2 Fig. Simulated gaze effects in a two-alternative choice task.** Models were simulated 500 times for each combination of Q-value magnitude, ranging from 0.1 to 0.9 in increments of 0.01, and proportional gaze allocated to the second alternative (gaze2), ranging from 0 to 1 in increments of 0.1. Both options were given the same Q-value in the simulations (i.e., one of the magnitudes listed above). The y-axis in each panel is the proportion of times the second alternative was chosen (prop2). The models without gaze data show no gaze effects. The Q * gaze and softmax(Q * gaze) models exhibit stronger gaze effects when the options have larger Q-values. The Q+gaze model, in contrast, exhibits a slightly stronger gaze effect when the Q-values are smaller. Models were simulated using the mean parameter estimates from Experiment 1.
(PNG)

**S3 Fig. Model fit for an example participant.** Trial-to-trial Q-values, proportional gaze data, and mean drift rates for participant 32 in the learning phase of Experiment 2 (softmax(Q+gaze) model parameter estimates: $\alpha$ = .15, $w_{rel}$ = .66, $\beta_Q$ = 0.33, $\beta_{gaze}$ = 1.03, $\theta$ = 2.06, $A$ = 151.31, $b$ = 281.12, $t_0$ = 77.71). The background colors indicate the participant's actual choice (green = correct option, red = incorrect option). The black circles represent the correct option on each trial; the black x's represent the incorrect option. The model correctly predicted 112 out of 120 (93%) of this participant's choices (in the mean drift rate plot, the red 0's indicate inaccurate predictions). Because the learning contexts were randomly ordered, the correct and incorrect options do not refer to the same underlying symbols on every trial.
(PNG)

**S4 Fig. Parameter recovery in Experiment 1.** The "softmax(Q) + gaze" and "softmax(Q + gaze)" models were simulated 100 times in the task from the first experiment (60 trials) using parameter values drawn from the prior distributions (see main text, *Materials and methods*). Then, the models were fit to the 100 simulated data sets to assess their ability to recover the true, data-generating parameters. Relationships between the generating and recovered parameters are shown with regression lines overlaid. The "softmax(Q) + gaze" model had the lowest accumulative one-step-ahead prediction error in Experiment 1, averaged across participants.
(PNG)

**S5 Fig. Parameter recovery in Experiment 2.** The "softmax(Q) + gaze" and "softmax(Q + gaze)" models were simulated 100 times in the task from the second experiment (232 trials) using parameter values drawn from the prior distributions (see main text, Materials and methods). Then, the models were fit to the 100 simulated data sets to assess their ability to recover the true, data-generating parameters. Relationships between the generating and recovered parameters are shown with regression lines overlaid. The "softmax(Q + gaze)" model had the lowest accumulative one-step-ahead prediction error in Experiment 2, averaged across participants.
(PNG)

**S6 Fig. Model fit to the aggregate choice-RT patterns in Experiment 1.** Proportion of correct choices and mean RT across learning trials. Error ribbons represent ±1 standard error.
(PNG)

**S7 Fig. Model fit to the aggregate choice-RT patterns in the learning phase of Experiment 2.** Proportion of correct choices and mean RT across learning trials. Error ribbons represent ±1 standard error. The purple lines show the fit of the original model with a static decision threshold. The orange lines show the fit of the modified model with a trial-dependent (decreasing) decision threshold, which was better at capturing the steep, nonlinear decrease in mean RT across the learning phase.
(PNG)

**S8 Fig. Model fit to individual choice accuracies and mean RTs in Experiment 1.**
(PNG)

**S9 Fig. Model fit to individual choice accuracies and mean RTs in the learning phase of Experiment 2.**
(PNG)

**S10 Fig. Model fit to the response time distributions in Experiment 1.** RT distributions for correct (maximizing) and incorrect (nonmaximizing) choices were pooled across participants. Incorrect RTs are negative for visualization purposes.
(PNG)

**S11 Fig. Model fit to the response time distributions in Experiment 2.** RT distributions for correct (maximizing) and incorrect (nonmaximizing) choices were pooled across participants. Incorrect RTs are negative for visualization purposes.
(PNG)

**S12 Fig. Model fit to the aggregate choice-RT patterns in the transfer test of Experiment 2.** Mean choice accuracy and mean RT as a function of trial type (C = congruent, N++ = neutral with both options having high relative values, N-- = neutral with both options having low relative values, I = incongruent). Error bars represent ±1 standard error.
(PNG)

**S13 Fig. Model fit to the gaze effects in Experiment 1.** Mean choice accuracy (top) and mean RT (bottom) as a function of the excess proportional gaze on the correct option, broken up into five quintiles. The quintiles were constructed by taking the difference between the proportional gaze for the correct (higher valued) and incorrect (lower valued) symbols on each trial, sorting the difference scores, and dividing them into five equal-sized bins, separately for each participant. The higher the quintile, the longer the correct option was fixated relative to the incorrect option. Error bars represent ±1 standard error.
(PNG)

**S14 Fig. Model fit to the gaze effects in the learning phase of Experiment 2.** Mean choice accuracy (top) and mean RT (bottom) as a function of the excess proportional gaze on the correct option, broken up into five quintiles. The quintiles were constructed by taking the difference between the proportional gaze for the correct (higher valued) and incorrect (lower valued) symbols on each trial, sorting the difference scores, and dividing them into five equal-sized bins, separately for each participant. The higher the quintile, the longer the correct option was fixated relative to the incorrect option. Error bars represent ±1 standard error.
(PNG)

**S15 Fig. Model fit to the gaze effects in the transfer test of Experiment 2.** Mean choice accuracy (top) and mean RT (bottom) as a function of the excess proportional gaze on the correct option, broken up into five quintiles. The quintiles were constructed by taking the difference between the proportional gaze for the correct (higher valued) and incorrect

(lower valued) symbols on each trial, sorting the difference scores, and dividing them into five equal-sized bins, separately for each participant. The higher the quintile, the longer the correct option was fixated relative to the incorrect option. Error bars represent ±1 standard error.
(PNG)

**S1 Table. Multiple regression predicting individual choice accuracy from RL-SSM parameters (Experiment 1).**
(PDF)

**S2 Table. Multiple regression predicting individual choice accuracy from RL-SSM parameters (Experiment 2: Learning phase).**
(PDF)

**S3 Table. Multiple regression predicting individual mean RT from RL-SSM parameters (Experiment 1).**
(PDF)

**S4 Table. Multiple regression predicting individual mean RT from RL-SSM parameters (Experiment 2: Learning phase).**
(PDF)

**S5 Table. Multiple regression predicting incongruent trial accuracy from RL-SSM parameters (Experiment 2: Transfer test).**
(PDF)

**S6 Table. Linear mixed-effects model predicting proportional gaze advantage for the correct option from trial number, EV difference and overall expected value in Experiment 1.**
(PDF)

**S7 Table. Linear mixed-effects model predicting proportional gaze advantage for the correct option from trial number and overall expected value in the learning phase of Experiment 2.**
(PDF)

**S8 Table. Logistic mixed-effects model predicting choice accuracy from trial number, EV difference, overall EV, and proportional gaze advantage for the correct option in Experiment 1.**
(PDF)

**S9 Table. Logistic mixed-effects model predicting choice accuracy from trial number, overall EV, and proportional gaze advantage for the correct option in the learning phase of Experiment 2.**
(PDF)

**S10 Table. Linear mixed-effects model predicting log RT from trial number, EV difference, overall EV, and proportional gaze advantage for the correct option in Experiment 1.**
(PDF)

**S11 Table. Linear mixed-effects model predicting log RT from trial number, overall EV, and proportional gaze advantage for the correct option in the learning phase of Experiment 2.**
(PDF)

**S12 Table. Logistic mixed-effects model predicting choice accuracy from EV difference, relative value difference, overall EV, overall relative value, and proportional gaze advantage for the correct option in the transfer test of Experiment 2.**
(PDF)

**S13 Table. Linear mixed-effects model predicting log RT from EV difference, unsigned relative value difference, unsigned proportional gaze difference, overall expected value and overall relative value in the transfer test of Experiment 2.**
(PDF)

## Acknowledgments

The authors thank Andrew Dolinsky, Rishi Heggawadi, Nadiah Layne, Trevor Rosenthal, Joelle Sacks, and Elaine Yu for their assistance with data collection, and an anonymous reviewer for suggesting the "softmax(Q + gaze)" model.

## Author contributions

**Conceptualization:** William M. Hayes.

**Data curation:** William M. Hayes.

**Formal analysis:** William M. Hayes, Melanie J. Touchard.

**Investigation:** William M. Hayes, Melanie J. Touchard.

**Methodology:** William M. Hayes.

**Software:** William M. Hayes.

**Supervision:** William M. Hayes.

**Visualization:** William M. Hayes.

**Writing – original draft:** William M. Hayes, Melanie J. Touchard.

**Writing – review & editing:** William M. Hayes, Melanie J. Touchard.

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
