## [Decision Letter · Decision Letter 0]

8 Dec 2025

A reinforcement learning and sequential sampling model constrained by gaze data

PLOS Computational Biology

Dear Dr. Hayes,

Thank you for submitting your manuscript to PLOS Computational Biology. After careful consideration, we feel that it has merit but does not fully meet PLOS Computational Biology's publication criteria as it currently stands. Therefore, we invite you to submit a revised version of the manuscript that addresses the points raised during the review process.

We look forward to receiving your revised manuscript.

Kind regards,

Bastien Blain

Academic Editor

PLOS Computational Biology

Marieke van Vugt

Section Editor

PLOS Computational Biology

**Journal Requirements:**

2) Some material included in your submission may be copyrighted. According to PLOSu2019s copyright policy, authors who use figures or other material (e.g., graphics, clipart, maps) from another author or copyright holder must demonstrate or obtain permission to publish this material under the Creative Commons Attribution 4.0 International (CC BY 4.0) License used by PLOS journals. Please closely review the details of PLOSu2019s copyright requirements here: PLOS Licenses and Copyright. If you need to request permissions from a copyright holder, you may use PLOS's Copyright Content Permission form.

Potential Copyright Issues:

i) Figure 1B. Please confirm whether you drew the images / clip-art within the figure panels by hand. If you did not draw the images, please provide (a) a link to the source of the images or icons and their license / terms of use; or (b) written permission from the copyright holder to publish the images or icons under our CC BY 4.0 license. Alternatively, you may replace the images with open source alternatives. See these open source resources you may use to replace images / clip-art:

**Reviewers' comments:**

Reviewer's Responses to Questions

**Comments to the Authors:**

Reviewer #1: This interesting article investigates the links between gaze and choice in reinforcement learning (RL) paradigms, through the lens of sequential sampling models (SSM), specifically linear ballistic accumulator (LBA) models. The authors examine two RL experiments and compare 7 different models. They conclude that in both experiments, the model with the softmax function and additive gaze effect performs best overall.

Overall, my impression of the paper is positive. Using SSMs to better understand RL behavior seems very promising, and accounting for gaze effects is a clearly important step. It’s great that there are multiple datasets, since behavior in RL tasks may vary quite a lot depending on the setup. I appreciate that the authors test several different models, that they check parameter recovery for their preferred model, and that they provide extensive analyses of the relative goodness of fits of the models in the supplements. This allows the reader to thoroughly evaluate the claims and draw their own conclusions.

However, I do have some concerns about the paper, mostly regarding the modeling.

First, it strikes me as problematic to have set the theta parameter in the softmax function to 50. The optimal value of this parameter depends on the units of the values and so must surely also depend on the experiment, the participant, etc. I worry that setting this parameter to different values could affect the rankings of the sotfmax models. This concern is reinforced by the substantial difference between softmax(Q*gaze) and softmax(Q)*gaze in terms of how they explain the RT curves in Figs. S6 and S13-15. The authors should either fit theta for the model comparison exercise or demonstrate that the ranking are insensitive to a range of theta values. Once they have a preferred model, then they could fix theta to estimate the rest of the parameters, assuming that parameter recovery is negatively affected by letting theta be free.

Second, related to the previous point, I’m concerned by Figure 2A. Given that softmax(Q) + gaze is the best model in both experiments, shouldn’t we expect a relatively similar ranking of the other models across the two experiments? Aside from Q + gaze being the second best model, the ranking of the other models looks uncorrelated between the experiments. This deepens my concern that the model fits may be unduly influenced by the choice of theta.

Third, I didn’t understand why the authors chose to induce competition in their model using the softmax function rather than using value difference between options. To their credit, in the Discussion they do mention that these models aren’t really comparable to the DDM or aDDM, but it’s unclear why they chose this modeling approach that precludes such comparisons. At the very least, this deserves more discussion, if not fitting models that have value differences in the drift rates. On a related note - why do the non-softmax models yield such flat RT curves relative to the data? Presumably this is because of the lack of competition in the model. It seems like those non-softmax models require some form of competition to have a fighting chance in these comparisons.

Fourth, there’s very little discussion of the starting point parameters, how we should interpret them in these experiments, and how we should interpret their results.

Fifth, was the Experiment 2 model fit to only the training data, or also to the transfer data? This wasn’t clear from the Methods.

Sixth, why is there no softmax(Q+gaze) model in the comparison? That seems like an obvious omission given that there are two multiplicative softmax models.

I also have some other questions/concerns:

Regarding the literature, the authors neglect to mention Westbrook et al. 2020, which is an important paper, as it seemingly resolves the debate between the additive and multiplicative gaze models. In particular, they argue that early in the decision gaze is multiplicative, but later, once the decision has been tentatively made, gaze becomes additive. This seems important to mention here, as the authors’ modeling approach doesn’t allow for a hybrid model that switches from multiplicative to additive over time. But perhaps they could include a model that has both additive and multiplicative components.

Do the authors also have eye-tracking data during the feedback phase in Experiment 1? It would be useful to know whether participants actually look at all four stimuli during feedback. If participants only look at the target, or chosen, options, that might suggest a different model with only partial learning.

Do the authors have an explanation for why gaze has a larger effect on choice in Experiment 1?

I think the authors are somewhat burying the lede by not making a bigger deal about the binomial distribution of relative vs. absolute value representation. It seems like that is an important finding and should be mentioned in the abstract.

It seems worth noting that in the aDDM, faster RT across trials and for N++ trials (and slow N— trials) would be automatically explained by the multiplicative gaze effects, as the Q values are higher in those cases and the aDDM predicts faster decisions for higher overall value.

In the results on pages 10 and 11, what effect does overall value have on choice and RT?

Do the authors have an explanation for why the trial interaction is different between the two experiments?

The fitted non-decision times are substantially shorter than we normally see in other studies. This seems worth commenting on.

In the declining threshold model, is delta significantly different from 0?

The authors may want to comment on Fig S12, namely the fact that the non-softmax models seem to do a better job of capturing the RT changes for N++ and N- -.

References:

Westbrook, A., Van Den Bosch, R., Määttä, J. I., Hofmans, L., Papadopetraki, D., Cools, R., & Frank, M. J. (2020). Dopamine promotes cognitive effort by biasing the benefits versus costs of cognitive work. Science, 367(6484), 1362-1366.

Reviewer #2: This is a very good manuscript, clearly and thoroughly presenting a well-designed study. The study shows how visual attention to choice options impacts the learning (or perhaps elucidating) of the values of the options, in addition to the previously-reported impact on the choices themselves. The authors present a computational model that includes elements of reinforcement learning, sequential sampling, and gaze bias, and their model successfully accounts for a variety of experimental findings. I think this paper is already well along the way towards being publishable. I list below some minor suggestions and questions for the authors to consider:

lines 141-145: Perhaps the reader would benefit from a bit of an explanation of this, specifically how the softmax rule makes the decision variables dependent on each other and thus the racing accumulators are not independent.

255-257: The authors should note that it comparing drift scalar components should be done in a relative way, not absolute. Thus the statement should perhaps be about gaze having a stronger influence on choice *relative* to the influence of learned values.

377-379: The authors show that gaze difference has an effect on choice even after controlling for other factors, such as value difference. However, they do so by including all the factors as co-regressors in the same regression model. A stricter way of controlling for the influence of one variable would be to orthogonalize the second variable with respect to the first (e.g., regressing one on the other and taking the residuals as the second regressor in the model of interest). Did the authors try that, and if so, did it change any of the results?

507-509: I wonder if allowing the softmax inverse temperature parameter to be free would be one way of interpolating between a linear mapping and the softmax mapping that the authors used (with a fixed inverse temperature).

Reviewer #3: The manuscript introduces a reinforcement-learning sequential-sampling model (RL-SSM) that integrates eye-tracking data to predict choices and response times in repeated decision tasks. The model assumes that subjective values (Q-values) learned through trial-and-error and proportional gaze independently contribute to drift rate in the sequential sampling process. Two experiments evaluate the model against alternatives that differ in how gaze and values are combined. The authors find that a softmax-based value mapping with an additive gaze term provides the best out-of-sample predictions. The model captures a wide range of behavioral and gaze-related effects, and the work addresses an important gap in the literature by combining learning and gaze-influenced decision dynamics within a computationally tractable model.

Overall, the manuscript presents a valuable integration of gaze into RL-SSMs and offers compelling empirical demonstrations. The approach is promising and clearly articulated. Several points would, however, benefit from clarification or extension to strengthen the interpretation and robustness of the findings.

Major Comments

1. The model comparison relies exclusively on APE, which, as I understand it, does not penalize model complexity. Since the winning model is also the most flexible, a complexity-adjusted comparison or a brief discussion of how complexity affects APE would help ensure that the conclusions are not driven by differences in the number of free parameters.

2. In Experiment 1, fixing the relative-encoding parameter ω_rel to 0 is understandable given identifiability constraints, but it is not clear why absolute encoding is preferred over fully relative encoding. Since many participants in Experiment 2 show strong relative valuation, comparing fixed ω_rel=0 versus ω_rel=1, or providing a principled justification for choosing ω_rel=0, would help validate this modeling decision.

3. [238-248] The trial-by-trial example in Fig. 2B is helpful for illustrating how gaze can improve predictions on erroneous trials. However, since this point is later referenced in a more general way in the discussion, a brief group-level summary of how often the gaze-augmented model improves error prediction relative to the no-gaze model would help clarify whether the effect is systematic across participants or limited to selected cases.

4. [253-257] The discussion of β_Q and β_gaze relies on visual inspection of their empirical distributions. If claims about positivity or parameter differences across experiments are intended to be inferential, providing confidence intervals or statistical tests would avoid possible overinterpretation.

5. [370-374] In the mixed-effects analyses of accuracy and RT, the model uses fixed task-defined EVs as predictors. Because these EVs do not reflect learning-induced changes in subjective value differences, I am not sure I fully understand how this specification controls for learning effects when estimating the additional contribution of gaze. Clarifying this rationale, or considering learned Q-value differences, would strengthen the argument.

6. A trial-dependent decreasing decision threshold improves RT fits (S2 Appendix), yet this model is not included in the formal model comparison. A brief explanation of why it was not evaluated with APE would clarify whether this decision reflects theoretical priorities, identifiability constraints, or practical considerations.

7. [519-526] The discussion proposes future extensions in which models would predict gaze based on learned values and other factors (such as salience or uncertainty). In learning tasks, however, gaze may also influence value updating itself. Acknowledging this reciprocal influence would provide a more complete perspective on potential extensions of the modeling framework.

Minor

a. Fixing θ enhances identifiability, but the specific value of 50 is only briefly justified. Since θ strongly affects drift mapping, a brief sensitivity analysis or additional justification would increase confidence that results do not hinge on this value.

b. [139-140] The text states that the steep softmax “produces greater overall accuracy (S1 Fig.),” but S1 Fig. does not present accuracy results. Adjusting the citation or wording would help readers follow the argument more easily.

c. [118] In the notation “Uniform(0,A)”, A is not defined until line 190.

d. The main model line is described as “maroon” through the manuscript (e.g. [269]) but to me it appears more purple.

e. I did not find citations to Figures S10 and S11 or to Tables S6 to S13 in the main text.

**Have the authors made all data and (if applicable) computational code underlying the findings in their manuscript fully available?**

Reviewer #1: Yes

Reviewer #2: Yes

Reviewer #3: **No:** I don't know if it's available. It's not specified in the manuscript.

PLOS authors have the option to publish the peer review history of their article (what does this mean? ). If published, this will include your full peer review and any attached files.

**Do you want your identity to be public for this peer review?** For information about this choice, including consent withdrawal, please see our Privacy Policy .

Reviewer #1: No

Reviewer #2: No

Reviewer #3: No

**Figure resubmission:**

After uploading your figures to PLOS’s NAAS tool - https://ngplosjournals.pagemajik.ai/artanalysis, NAAS will process the files provided and display the results in the "Uploaded Files" section of the page as the processing is complete. If the uploaded figures meet our requirements (or NAAS is able to fix the files to meet our requirements), the figure will be marked as "fixed" above. If NAAS is unable to fix the files, a red "failed" label will appear above. When NAAS has confirmed that the figure files meet our requirements, please download the file via the download option, and include these NAAS processed figure files when submitting your revised manuscript
---

## [Decision Letter · Decision Letter 1]

23 Feb 2026

Dear Mr. Hayes,

We are pleased to inform you that your manuscript 'A reinforcement learning and sequential sampling model constrained by gaze data' has been provisionally accepted for publication in PLOS Computational Biology.

Best regards,

Bastien Blain

Academic Editor

PLOS Computational Biology

Marieke van Vugt

Section Editor

PLOS Computational Biology

Reviewer's Responses to Questions

**Comments to the Authors:**

Reviewer #1: The authors have addressed my concerns. Congratulations on a great paper.

Reviewer #2: The authors have satisfied my concerns with their responses and revisions.

Reviewer #3: The authors have made a clear and commendable effort to address my previous comments. The revisions clarify key methodological choices, strengthen the analyses, and improve the overall clarity of the manuscript.

I have no further substantive concerns and am pleased with the current version.

**Have the authors made all data and (if applicable) computational code underlying the findings in their manuscript fully available?**

Reviewer #1: Yes

Reviewer #2: Yes

Reviewer #3: Yes

PLOS authors have the option to publish the peer review history of their article (what does this mean? ). If published, this will include your full peer review and any attached files.

**Do you want your identity to be public for this peer review?** For information about this choice, including consent withdrawal, please see our Privacy Policy .

Reviewer #1: No

Reviewer #2: No

Reviewer #3: No

---

## [Editor Report · Acceptance letter]

PCOMPBIOL-D-25-01886R1

A reinforcement learning and sequential sampling model constrained by gaze data

Dear Dr Hayes,

I am pleased to inform you that your manuscript has been formally accepted for publication in PLOS Computational Biology. Your manuscript is now with our production department and you will be notified of the publication date in due course.

With kind regards,

Zsofia Freund
